# Comparison of the Effect of Native 1,4-Naphthoquinones Plumbagin, Menadione, and Lawsone on Viability, Redox Status, and Mitochondrial Functions of C6 Glioblastoma Cells

**DOI:** 10.3390/nu11061294

**Published:** 2019-06-07

**Authors:** Daiva Majiene, Jolita Kuseliauskyte, Arturas Stimbirys, Aiste Jekabsone

**Affiliations:** 1Laboratory of Biochemistry, Neuroscience Institute, Lithuanian University of Health Sciences, Eiveniu str. 4, LT-50162 Kaunas, Lithuania; 2Department of Drug Technology and Social Pharmacy, Lithuanian University of Health Sciences, Sukileliu st. 13, LT-50162 Kaunas, Lithuania; jolitkuseliausk@gmail.com; 3Department of Food Safety and Quality, Lithuanian University of Health Sciences, Tilzes st. 18, LT-47181 Kaunas, Lithuania; arturas.stimbirys@lsmuni.lt; 4Laboratory of Molecular Neurobiology, Neuroscience Institute, Lithuanian University of Health Sciences, Eiveniu str. 4, LT-50162 Kaunas, Lithuania; aiste.jekabsone@lsmuni.lt; 5Laboratory of Pharmaceutical Sciences, Institute of Pharmaceutical Technologies, Lithuanian University of Health Sciences, Sukileliu st. 13, LT-50162 Kaunas, Lithuania

**Keywords:** plumbagin, menadione, lawsone, reactive oxygen species, mitochondrial respiration, C6 glioma cell culture

## Abstract

Background: 1,4-naphthoquinones, especially juglone, are known for their anticancer activity. However, plumbagin, lawsone, and menadione have been less investigated for these properties. Therefore, we aimed to determine the effects of plumbagin, lawsone, and menadione on C6 glioblastoma cell viability, ROS production, and mitochondrial function. Methods: Cell viability was assessed spectrophotometrically using metabolic activity method, and by fluorescent Hoechst/propidium iodide nuclear staining. ROS generation was measured fluorometrically using DCFH-DA. Oxygen uptake rates were recorded by the high-resolution respirometer Oxygraph-2k. Results: Plumbagin and menadione displayed highly cytotoxic activity on C6 cells (IC_50_ is 7.7 ± 0.28 μM and 9.6 ± 0.75 μM, respectively) and caused cell death by necrosis. Additionally, they increased the amount of intracellular ROS in a concentration-dependent manner. Moreover, even at very small concentrations (1–3 µM), these compounds significantly uncoupled mitochondrial oxidation from phosphorylation impairing energy production in cells. Lawsone had significantly lower viability decreasing and mitochondria-uncoupling effect, and exerted strong antioxidant activity. Conclusions: Plumbagin and menadione exhibit strong prooxidant, mitochondrial oxidative phosphorylation uncoupling and cytotoxic activity. In contrast, lawsone demonstrates a moderate effect on C6 cell viability and mitochondrial functions, and possesses strong antioxidant properties.

## 1. Introduction

Glioblastoma multiforme (GBM) is the most aggressive, invasive, and undifferentiated type of central nervous system tumor in adults [1]. Although several treatment options are available, including surgery along with adjuvant chemo- and radio-therapy, the disease has a poor prognosis, with a median survival rate of 12–17 months. GBM cells are characterized by an increased resistance to most of antitumor drugs. The main compound used for treatment of the disease is temozolomide, but the efficiency of the drug is limited and the treatment is accompanied by numerous side effects [2]. To improve the survival of patients, there is an intensive effort to design new, more efficient chemotherapeutic agents and biological therapies, such as immunotherapy and stem cell therapy, as well as better diagnostics and prediction of GBM. The invention of preventive strategies, such as personalized vaccines, against GBM has also started and the initial human trials have already shown a better survival rate [3]. However, despite advances in GBM treatment, the malignancy is still the most challenging task in clinical oncology. There is evidence that alternative medicine measures such as the application of natural biologically-active substances with anticancer activity could improve GBM treatment results. Therefore, it is important to investigate natural biologically-active compounds that can be used for cancer prevention and treatment, both alone and in combination with the conventional anticancer therapy.

Quinones, and especially 1,4-naphthoquinone compounds, are wide-spread phenolic compounds in nature. They are products of bacterial and fungal, as well as high-plant, secondary metabolism and perform cytoprotective functions. Naphthoquinone-rich plant extracts are well-known antibacterial and anti-inflammatory preparations in folk medicine worldwide [4].

During recent decades, a growing number of studies have shown an increasing interest in purified biologically-active 1,4-naphthoquinones. We and other research groups have shown that juglone exerts cytotoxic effect by increasing intracellular reactive oxygen species (ROS) generation and modulating mitochondrial oxidative phosphorylation. Therefore, juglone could be a promising candidate for GBM prevention and treatment [5,6]. However, the effect of other structurally similar compounds, including lawsone (Lw), plumbagin (PL), and menadione (Mn) (Figure 1) against GBM cells is not so well-defined.

Plumbagin is a yellow pigmented secondary metabolite usually isolated from the roots of *Plumbaginaceae, Ancestrocladaceae*, and *Dioncophyllaceae* families. The compound possess multiple biological properties, including those that are antiviral, antibacterial, antiprotozoal, and anti-inflammatory [7]. Some studies have already confirmed the antitumor effect of plumbagin. It prevented the proliferation and survival of prostate cancer cell lines DU145 and PC-3, human endocrine-resistant breast cancer, and other tumor cells [8,9]. One of the mechanisms of plumbagin-induced cytotoxicity is related to the prooxidant activity of this compound. Reactive oxygen species generation in cells correlates with the intensity of apoptosis and directly depends on the plumbagin concentration [10].

Menadione belongs to the vitamin K family and is known as vitamin K3 [11]. Vitamin K generally maintains healthy blood clotting and prevents excessive bleeding and hemorrhage, and is also important for maintaining healthy bone structure and for carbohydrate storage in the body [12]. There is evidence that menadione inhibits growth and induces apoptosis in several tumor cell types, such as breast, liver, and nasopharyngeal carcinomas, in vitro and in rat models [13]. Menadione-induced cytotoxicity involves various mechanisms, including oxidative damage induced by ROS. It has also been shown that menadione is less toxic to non-cancer cells. This makes it a promising candidate as a nutritional chemopreventive, as well as a potential chemotherapeutic compound to treat tumors [14].

Lawsone is a red-orange pigment present in henna *Lawsonia inermis L*. leaves. The plant extracts and purified constituent of henna are known as cosmetic and medicinal agents. In folk medicine, it is used for a variety of activities, including antibacterial, antifungal, antioxidant, and anti-inflammatory [15]. In vitro studies on different cancer cell lines have shown that lawsone inhibited growth of HCT-15 human colon cancer cells by blocking the S-phase of the cell cycle [16]. However, there are only a very few studies about the anticancer activity of naturally-derived lawsone. Most research is performed on synthetic derivatives of lawsone, and the studies demonstrate significant anticancer effect [17,18].

The aim of this study was to determine and compare the effects of plumbagin, lawsone, and menadione on viability of glioblastoma C6 cells, generation of intracellular ROS, and modulation of mitochondrial function.

## 2. Materials and Methods

### 2.1. Chemicals and Reagents

All solvents, reagents, and standards used in this study were of analytical grade. 

Plumbagin, menadione, lawsone (>97% purity), and 3-(4,5-dim- ethydiazol-2-yl)-2,5-diphenyltetrazolium bromide were purchased from Sigma-Aldrich (st.Louis, MO, USA). All cell culture reagents were obtained from Gibco (Fisher Scientific, UK).

### 2.2. Cell Line and Cell Culture

Rat glioma C6 cells were purchased from the Cell Lines Service GmbH (Germany). C6 cells were seeded in culture flasks containing Dulbecco’s Modified Eagle Medium (DMEM) with 10% fetal bovine serum, 100 U/mL penicillin, and 100 µg/mL streptomycin. The cultures were then incubated at 37 °C, with 5% CO_2_ and saturated humidity. Additionally, 24 h prior to treatment with plumbagin, menadione, or lawsone, the cells were transferred to a 96-well plate at a density of 20,000 cells/well.

### 2.3. Cell Viability Assessment

The viability of cells was assessed by measuring their ability to metabolize 3-(4,5-Dimethylthiazol-2-yl)-2,5-Diphenyltetrazolium Bromide (MTT) (as described in [5]) and by a nuclear fluorescent staining assay. In the first experimental series, C6 cells were treated with different concentrations of either plumbagin, menadione, or lawsone for 24 h. Concentrations for plumbagin and menadione were 1 µM, 3 µM, 5 µM, 8 µM, 10 µM, 12.5 µM, 15 µM, 20 µM, and 25 µM and for lawsone were 10 µM, 25 µM, 50 µM, 75 µM, 100 µM, 250 µM, 500 µM, 750 µM, and 1000 µM. In experiments with ascorbate, C6 cells were treated with 5 µM and 15 µM plumbagin, together with 100 µM and 250 µM of ascorbic acid for 24 h. After incubation with 1, 4-naphthoquinones, the cells were double-stained with Hoechst 33,342 (15 µg/mL) and propidium iodide (PI; 5 µg/mL) for 15 min, and the viability was assessed under the fluorescence microscope OLYMPUS IX71SIF-3 (Olympus Corporation, Tokyo, Japan). IC_50_ was calculated by SigmaPlot 13.0 version (Systat Software Inc., San Jose, CA, USA) software by means of the four-parameter logistic function. 

### 2.4. Measurement of Intracellular ROS Concentration

Intracellular ROS were assessed using the 2′,7′-dichlorofluorescein diacetate (DCFH-DA). After the incubation of C6 cells in 96-well plates for 24 h, they were incubated with DCFH-DA (10 µM) in Hank’s Balanced Salt Solution (HBSS) at 37 °C for 30 min. During this time, a part of DCFH was diffused into the cells. The excess dye was washed twice with phosphate-buffered saline (PBS). Wells were filled with an HBSS and different concentrations of plumbagin and menadione (1–25 µM) and lawsone (10–1000 µM) were added. In the presence of cellular oxidizing agents, DCFH is oxidized to the highly fluorescent compound dichlorofluorescein, so the fluorescence intensity is proportional to the amount of ROS produced in the cells. The fluorescence of dichlorofluorescein was detected by a fluorometer at excitation and emission wavelengths of 488 and 525 nm, respectively.

The control level of intracellular ROS was determined using appropriate amounts of solvents and the fluorescence intensity in control samples at time point “0” was assumed to be 100%.

### 2.5. Assessment of Mitochondrial Oxygen Consumption

The oxygen uptake rates of mitochondria in C6 cells were recorded using high-resolution respirometer Oxygraph-2k (Oroboros Instruments, Innsbruck, Austria) as described in [19]. The measurement medium contained 0.5 mM EGTA, 3 mM MgCl_2_, 60 mM K-lactobionate, 20 mM Taurine, 10 mM KH_2_PO_4_, 20 mM HEPES, and 110 mM sucrose (pH 7.1 at 37 °C). At the beginning of the experiment, C6 cells were placed into the respirometric chamber and permeabilized with digitonin (10 µg/mL). After cell permeabilization, the non-phosphorylating respiration rate (V_L_) was measured with the mitochondrial I complex substrates glutamate (5 mM) and malate (5 mM), and the mitochondrial II complex substrate succinate (12 mM). The uncoupled respiration rate was measured using 10–50 μM of 2,4-Dinitrophenol (DNF). For an investigation of the effect of compounds on the non-phosphorylating respiration rate, 0.5–3 µM of plumbagin and menadione, and 10–200 µM of lawsone were added into the chamber and the respiration rate was recorded (Appendix A).

After recording the non-phosphorylating respiration rate, the adenosine diphosphate (ADP, 1 mM) was added, and the maximal rate of oxidative phosphorylation (VADP) was recorded. For investigation of the effect of compounds on VADP, previously mentioned amounts of plumbagin, lawsone, and menadione solutions were added and the respiration rate was measured (Appendix A).

Cell mitochondrial respiration rates were expressed as nmol O / s × 10^6^ cells × mL.

Control experiments were carried out by registering mitochondrial respiration parameters using solvents only. Solvent amounts were equal to those of added solutions. There were no changes in mitochondrial respiration rates after the addition of solvents alone.

### 2.6. Statistical Analysis

The results are presented as means ± standard error. Statistical analysis was performed by one-way analysis of variance (ANOVA), followed by Dunnett’s post-test using the software package SigmaPlot 13.0 version (Systat Software Inc., San Jose, CA, USA). The value of *p* < 0.05 was taken as the level of significance.

## 3. Results

### 3.1. The Effect of 1, 4-Naphthoquinones on C6 Cell Viability

The treatment of C6 cells with plumbagin and menadione showed a dose-dependent decrease in cell viability (Figure 2). At the lowest concentrations (1–3 μM), the effect of plumbagin on cell viability was negligible (Figure 2A). After treatment with 5–12.5 μM plumbagin, the viability dropped from 65% to 26,5%. A total of 15–20 μM plumbagin further increased cell death up to nearly 100%. The fluorescent images confirmed the results of the MTT assay: plumbagin caused C6 cell death by necrosis in a concentration-dependent manner, as identified by red propidium iodide fluorescence. The calculated IC_50_ for plumbagin was 7.7 ± 0.28 μM.

The treatment of C6 cells with menadione revealed similar effects to the case of plumbagin (Figure 2B). Menadione at the lowest concentrations (1–3 μM) had no effect on cell viability. Increasing the concentration of the compound to 5–8 μM induced a viability drop to 20–49%. Menadione applied at the highest concentrations (20–25 μM) further decreased cell viability until few to no cells were left alive. The fluorescent images demonstrate that menadione caused C6 cell death by necrosis. The IC_50_ calculated for menadione was 9.6 ± 0.75 μM.

In contrast to plumbagin and menadione, lawsone applied at the concentration range of 10–100 μM had no statistically significant effect on C6 viability (Figure 2C). A total of 250 μM–1 mM lawsone decreased the cell viability to 18–45%. The fluorescent images obtained confirmed the results of the MTT assay: lawsone up to 1 mM only induced necrosis in less than half of the cells.

### 3.2. The Effect of 1,4-Naphthoquinones on the Intracellular ROS Concentration in a C6 Cell Culture

The changes in intracellular ROS generation were analyzed by an assessment of the fluorescence of oxidized DCFH that was loaded on cells before treatment with 1,4-naphthoquinones. After the incubation of C6 cells with 1 and 3 μM plumbagin for up to 3 h, the level of fluorescence remained similar to that of control cells (Figure 3A). A total of 5–12.5 μM plumbagin did not induce significant changes in cell fluorescence for up to 1 h of treatment. However, when prolonging the incubation with plumbagin at this concentration to 1.5–3 h, the amount of intracellular ROS increased to 12–55%. Furthermore, 15 μM and 18 μM plumbagin statistically significantly increased cell fluorescence in all incubation periods tested in this study. The maximal stimulation of ROS production was found after 3 h: 15 μM plumbagin induced a 68% increase, and 18 μM plumbagin resulted in a 76% increase compared to the untreated control.

Up to 5 μM menadione did not induce a significant increase in ROS production after any time period evaluated in the study (Figure 3B). A total of 8–15 μM menadione statistically significantly elevated the intracellular ROS amount after 1.5–3 h of incubation. After 3 h, menadione at this concentration range increased the ROS level to 16–50%. The highest concentrations (20–25 μM) of menadione increased the fluorescence intensity in cells after 30 min of incubation, and this effect increased with the incubation time, reaching 69% for 20 μM menadione and 100% for 25 μM menadione after 3 h, respectively.

In contrast to plumbagin and menadione, lawsone at all concentrations decreased ROS-related fluorescence intensity in C6 cells (Figure 3C). A total of 25 μM lawsone significantly lowered ROS-induced fluorescence after 0.5 h incubation. Additionally, 50 μM and higher lawsone reduced the intracellular fluorescence immediately after addition. The results indicate that 50–1000 μM lawsone has intracellular ROS-decreasing activity.

### 3.3. Cytotoxic Effect of Plumbagin was Partially Prevented by Ascorbate

The results previously obtained in this study revealed that the extent of cytotoxicity of 1,4-naphthoquinones correlates with the amount of ROS generated by these compounds, suggesting that ROS might be the key triggers of C6 cell death. To test this hypothesis, we treated C6 cells with plumbagin, which revealed the most potent oxidative and cytotoxic properties (correlation coefficient r = −0.99), together with a classical antioxidant ascorbate. The antioxidant treatment slightly (by 10–14%) prevented cell death induced by 5 µM plumbagin, but did not affect cell death induced by 15 µM plumbagin (Figure 4).

### 3.4. The Effect of 1,4-Naphthoquinones on the Mitochondrial Respiration Rate in C6 Cells

There are data showing that anticancer properties of the 1,4-naphthoquinone juglone are related to the modulation of mitochondrial activity [6]. Therefore, in this study, we tested how plumbagin, menadione, and lawsone affect mitochondrial functions in C6 cells. Mitochondrial respiration was measured in a non-phosphorylating state, without the external addition of ADP, and in the presence of 1 mM ADP that causes the maximal stimulation of phosphorylation and subsequent activation of the respiratory chain. The non-phosphorylating respiration rate is named V_L_ (L for “leak”) because it is only caused by protons leaking across the inner mitochondrial membrane and is an indicator of the membrane permeability. The ADP-stimulated respiration rate is named V_ADP_ and provides information about the maximal respiratory capacity of the oxidative-phosphorylation system [20].

Plumbagin used at 0.5 µM concentration had a negligible effect on both V_L_ and V_ADP_ in C6 cells with mitochondrial complex I+II substrates glutamate + malate and succinate (Table 1). However, 1 µM plumbagin increased both respiration rates by 45% and 20%, respectively. After addition of plumbagin up to the final concentration of 3 µM, the mitochondrial respiration rate was further increased to the similar level as induced by complete uncoupling with DNP (Appendix A).

Menadione also increased V_L_ and V_ADP_ in a concentration-dependent manner (Table 1). A total of 1 µM menadione increased mitochondrial respiration by 38% and 33%. Additionally, 2 µM menadione caused ADP-stimulated mitochondrial respiration to increase by 70%, i.e., to the level of the maximal respiration capacity of uncoupled mitochondria. Furthermore, 3 µM menadione increased the mitochondrial non-phosphorylating respiration rate to a level similar to that achieved with the uncoupler.

A total of 10–20 µM lawsone had no significant effect on mitochondrial respiration. Moreover, 50–100 µM lawsone had a mild to moderate respiration stimulating effect; however, the maximal oxygen consumption capacity (respiration rate of uncoupled mitochondria) was not achieved, even at 200 µM (Table 1, Appendix A).

## 4. Discussion

In recent decades, studies of purified naphthoquinones have revealed that the compounds possess anticancer activity. Juglone is the best studied member of this group. The cytotoxic efficiency of juglone has been demonstrated in numerous malignancy models both in vitro and in vivo. One of the cytotoxic mechanisms initiated by juglone is related to ROS signaling pathway induction. It has also been shown that juglone suppresses mitochondrial complex I and decreases ATP synthesis [6]. Plumbagin, lawsone, and menadione are other abundantly found 1,4-naphthoquinones structurally similar to juglone. They are small molecules with lipophilic properties, therefore are expected to easily cross the blood–brain barrier and be applicable for GBM therapy.

Cell viability examination by both MTT and nuclear membrane integrity sensing fluorescence assays revealed that plumbagin and menadione efficiently kill GBM cells. This is in line with the findings of Li and other authors showing that plumbagin has a concentration-dependent proliferation suppressing and viability decreasing effect on human lung cancer cells A549 and H23 [21]. IC_50_ values calculated in this study were 10.87 μM for A549 cells and 7.80 μM for H23 cells. In other study, menadione was found to be toxic for pancreatic cancer cells, and IC_50_ established after 24 h treatment was 42.1 ± 3.5 μM [22]. Oztopcu et al. have shown the cytotoxic effect of menadione on C6 and human GBM cells, and declared IC_50_ = 41 μM for C6 cells and IC_50_ = 24 μM for human GBM cells [23]. Thus, the results of this study confirm the anticancer activity of plumbagin and menadione, even at lower concentrations than those previously reported. Furthermore, the efficient concentration ranges were comparable for plumbagin, menadione, and previously investigated juglone (IC_50_ value is 10.4 ± 1.6 μM) [5], as the compounds have a similar chemical structure. A nuclear fluorescent staining viability assay revealed that C6 cells affected by the three above mentioned naphthoquinones die via the necrotic pathway.

Lawsone is also structurally close to the above discussed compounds and some authors have found it to be cytotoxic for colorectal cancer cells HTC-15, with an IC_50_ = 12.5 μg/mL [16]. Wang et al. demonstrated that lawsone, when given to rats orally (200 mg/mL) for 8 weeks, suppressed the cell proliferation of colon tumors without affecting the cells of normal colon mucosa [24]. However, our results show that lawsone, even applied at a 1 mM concentration, had either no or a very small effect on C6 cell viability. The efficiency of lawsone in the animal study might be explained by the spontaneous as well as bacteria-mediated degradation of lawsone in gastrointestinal tract, resulting in a range of derivatives with potential anticancer properties [25]. Indeed, studies with naphthoquinones and other groups of polyphenol compounds indicate that they may be chemically unstable in gastrointestinal tract medium, and may also be decomposed by intestinal microflora [26]. Therefore, detailed in vivo stability and bioavailability studies and the mechanism of action of 1,4-naphthoquinone metabolites on gastrointestinal tissue represent an important area for future study. However, the different in vitro response of cancer cell cultures to lawsone points to some other causes, such as a cell type-specific response or different chemistry.

In order to define possible causes of such different actions between the three studied naphthoquinones, we have examined their effect on the generation of intracellular ROS. Treatment with higher than 5 μM plumbagin and higher than 8 μM menadione statistically significantly stimulated the production of intracellular ROS. These findings are similar to our earlier investigations with juglone [5] and to those of other authors. Eldhose and others have investigated the effect of plumbagin on human colorectal cancer cells HCT116 and found that this compound decreases cell viability in a concentration-dependent manner by stimulating generation of ROS that further interfere with the cell cycle in phase G1 and induce apoptosis [10]. Powolny and Singh have demonstrated that significant ROS production stimulation in human prostate cancer cells LNCaP is achieved after 2 h of incubation with 5 μM plumbagin [27]. Kim and co-authors also investigated changes in intracellular ROS levels in a menadione-affected ovary carcinoma cell line. They have established that 7.5 μM menadione after 24 h incubation increases the ROS amount more than twice and decreases the cellular viability by 50% compared to the control [28]. However, there are controversial studies showing the antioxidant activity of plumbagin and menadione. Kumar and others have demonstrated that menadione scavenged 23% of hydroxyl and 30% of superoxide radicals and was more effective in neutralizing these species compared to plumbagin and juglone at 120 μM and higher concentrations [29]. Wang and colleagues investigated the effects of plumbagin in rat models of myocardial infarction and revealed that 5 mg/kg plumbagin decreases the amount of ROS and fat peroxides in myocardial tissue [30]. Another research group has demonstrated that the antioxidant action of plumbagin can be even more potent than that of synthetic antioxidant dibutylhydroxytoluene [31]. In our study, we did not detect antioxidant properties of plumbagin and menadione. On the contrary, 50–1000 μM lawsone demonstrated statistically significant antioxidant activity. This was confirmed by other authors, who showed that lawsone possess free radical scavenging and metal chelating properties [15,32]. However, the study on rat erythrocytes performed by McMillan and co-authors revealed that 3 mM lawsone induced the generation of ROS including superoxide, yet the level of ROS produced was too small to deplete the glutathione pool in cells and induce oxidative stress [33]. The different effects of lawsone on ROS generation compared to the structurally similar compounds (plumbagin and menadione) could be determined by the -OH group at the C2 position [34]. It was shown that free hydroxyls and C2–C3 double bound in the B-ring are consistently associated with a higher antioxidant capacity in polyphenols [35]. Moreover, the presence of the -OH group, compared to the presence of -H and -CH_3_, also determines the stronger antioxidant properties of the compound [36].

It is known that mitochondria are one of the main ROS producers in the cell. Mitochondria use oxygen to make ATP under normal conditions, but upon exposure to harmful external or internal factors, for example, biologically active substances, mitochondria start to reduce molecular oxygen, incompletely generating superoxide radicals [37]. The production of mitochondrial ROS is directly dependent on mitochondrial function; thus, we have evaluated the effect of 1,4-naphthoquinones on mitochondrial oxygen consumption with mitochondrial complex I (glutamate + malate) and complex II (succinate) substrates. The results revealed that all investigated naphthoquinones have mitochondrial respiration-stimulating properties. Lawsone applied at 10–200 µM concentrations increased substrate-stimulated and oxidative phosphorylation rates, but did not completely uncouple oxidation from phosphorylation. It is known that mild uncoupling of oxidation and phosphorylation processes in mitochondria might be beneficial for cells because of the limitation of ROS production. Therefore, the antioxidant action of lawsone may occur via the direct neutralization of ROS and by modulating mitochondrial ROS production. Plumbagin and menadione significantly enhance both substrate-stimulated mitochondrial respiration and oxidative phosphorylation and completely uncouple oxidation from phosphorylation, thereby reducing ATP production. As a result of decreased ATP levels, necrotic cell death is initiated. It has been reported that agents interfering with electron transport, maintenance of the proton gradient, and the transfer of electrons to oxygen and ATP synthesis can be developed as cancer therapeutics [38]. Therefore, the strong uncoupling activity of plumbagin and menadione puts them in the line of such agents.

Our research confirmed that the viability-suppressing effects of plumbagin and menadione on the rat glioblastoma C6 cell line could be related to prooxidant and mitochondrial ATP-production efficiency-decreasing activity. Meanwhile, lawsone did not exert potent toxicity on C6 cells and stood out with strong antioxidative properties. This study complements the existing knowledge about the biological properties of 1,4-naphthoquinones and could be useful for further exploration of the activity mechanisms of these compounds and the differences between them.

## 5. Conclusions

1,4-naphthoquinones (plumbagin, menadione, and lawsone) decrease C6 cell viability in a concentration-dependent manner. Plumbagin and menadione exhibit high cytotoxic activity ((IC_50_ - 7.7μM and 9.6 μM, respectively), but lawsone has a significantly lower viability-decreasing efficiency.

1,4-naphthoquinones differ in their oxidative and oxidant-stimulating properties: 8 μM and higher menadione and plumbagin display prooxidant activity, whereas lawsone acts in an antioxidative manner and decreases intracellular ROS.

1,4-naphthoquinones have different effects on mitochondrial respiration. Lawsone mildly uncouples oxidation and phosphorylation processes. Plumbagin and menadione significantly enhance mitochondrial respiration and completely uncouple oxidation from phosphorylation, resulting in reduced ATP production.

## Figures and Tables

**Figure 1 nutrients-11-01294-f001:**
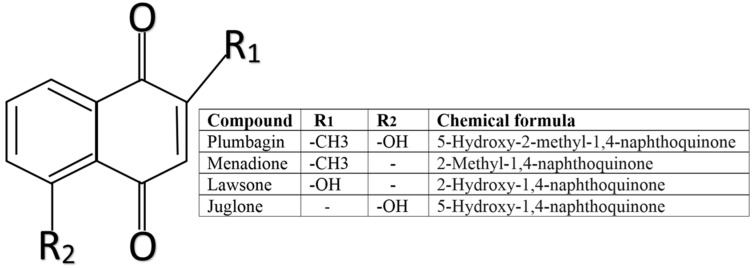
Structural formula of 1,4-naphthoquinones.

**Figure 2 nutrients-11-01294-f002:**
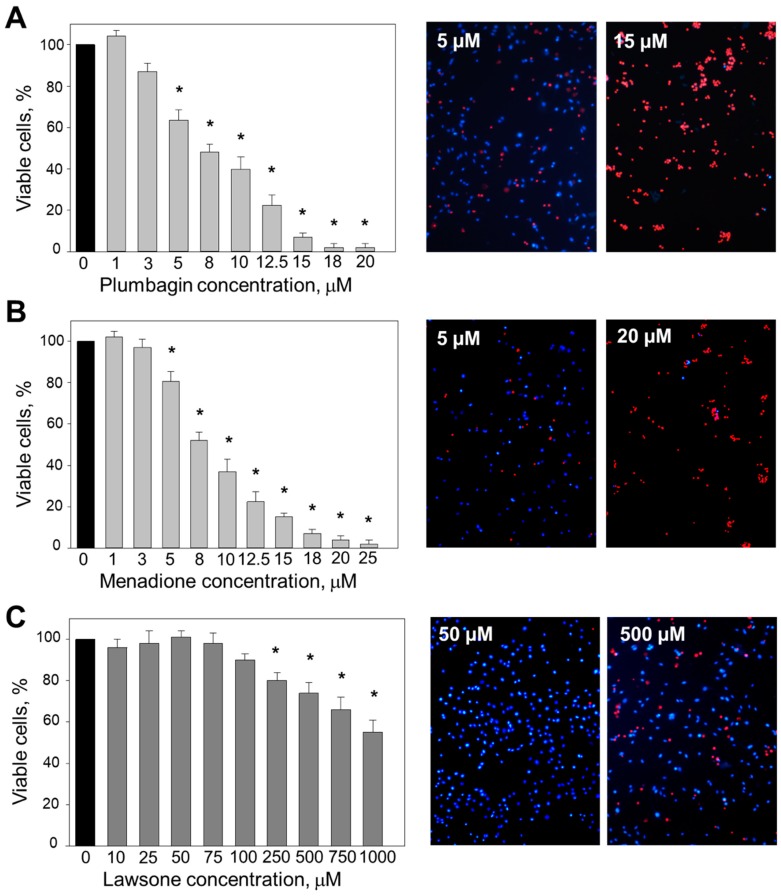
Effects of different concentrations of (**A**) plumbagin, (**B**) menadione, and (**C**) lawsone on the viability of C6 cells. Cell viability was assessed using (1) the MTT method and (2) double-staining with Hoechst 33342 and propidium iodide (PI). Hoechst 33342-positive cells, but lacking PI staining, were considered viable cells. PI-stained cells were considered necrotic. Data are presented as means of the percentage of the total cell number per micrograph ± SE (*n* = 5). * *p* < 0.05 versus control.

**Figure 3 nutrients-11-01294-f003:**
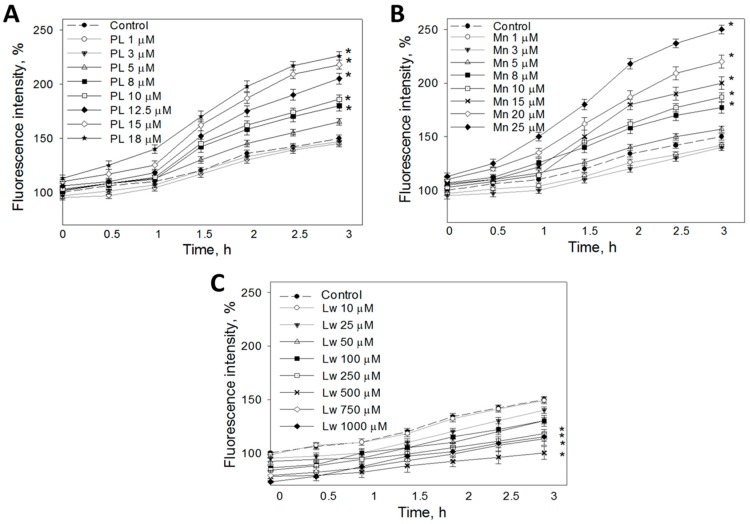
Effects of different concentrations of (**A**) plumbagin, (**B**) menadione, and (**C**) lawsone on intracellular ROS concentration. C6 cells were pre-treated with 10 μM DCFH-DA and then incubated with different concentrations of investigated compounds. Fluorescence intensity, which is proportional to the intracellular ROS concentration, was detected by a fluorometer at excitation and emission wavelengths of 544 and 590 nm, respectively. Data are presented as means of the percentage of control cells ± SE (*n* = 5). * *p* < 0.05 versus control.

**Figure 4 nutrients-11-01294-f004:**
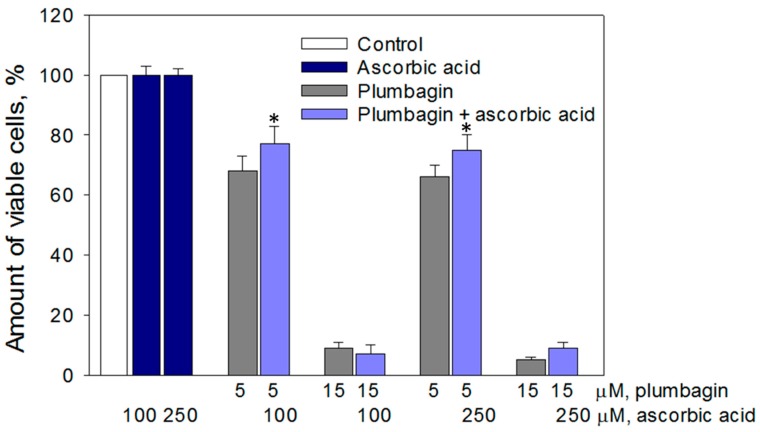
Effects of different concentrations of plumbagin and ascorbate on the viability of C6 cells. C6 cells were treated with 5 µM and 15 µM concentrations of plumbagin and 100 µM and 250 µM of ascorbate for 24 h. Cell viability was assessed using the MTT method (for details, see the Materials and Methods). Data are presented as means of the percentage of the untreated control cells ± SE (*n* = 3). * *p* < 0.05 versus control.

**Table 1 nutrients-11-01294-t001:** Effect of 1,4-naphthoquinones on oxygen consumption in mitochondria.

Treatment	µM	V_L_	V_DNF_	V_ADP_
Control		27.4 ± 5.2		58.4 ± 6.4
Control + DNF			86.5 ± 7.7	
Plumbagin	0.5	34.1 ± 5.6		62.9 ± 4.8
	1	39.8 ± 5.5 *		70.0 ± 5.4 *
	2	62.7 ± 3.9 *		77.9 ± 5.3 *
	3	84.5 ± 4.7 *		89.6 ± 7.2 *
Menadione	0.5	31.7 ± 3.1		60.8 ± 7.2
	1	37.8 ± 4.8 *		77.2 ± 6.8 *
	2	50.9 ± 5.7 *		90.7 ± 8.4 *
	3	84.1 ± 6.4 *		89.5 ± 7.8 *
Lawsone	10	30.3 ± 5.2		60.6 ± 4.8
	20	34.9 ± 4.7		63.1 ± 5.2
	50	43.6 ± 3.9 *		65.4 ± 3.9
	100	52.5 ± 4.3 *		68.1 ± 4.5 *
	200	61.3 ± 5.7 *		73.9 ± 5.3 *

Incubation of cells and measurement of the digitonin-permeabilized cell mitochondrial respiration were performed as described in Materials and Methods. V_L_ - non-phosphorylating mitochondrial respiration rate in the presence of 2 × 10^6^ cells/chamber and glutamate + malate (5 mM each) and succinate (12 mM); V_ADP_ - oxidative phosphorylation rate in the presence of respiratory substrates and 1 mM of ADP; V_DNF_ – uncoupled respiration rate in the presence 10–50 μM DNF; ***** – statistically significant effect as compared with the Control group.

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
