# Peer review of "Comparison of the Effect of Native 1,4-Naphthoquinones Plumbagin, Menadione, and Lawsone on Viability, Redox Status, and Mitochondrial Functions of C6 Glioblastoma Cells"

_nutrients, 2019, doi:10.3390/nu11061294_

Round 1

Reviewer 1 Report

The manuscript on viability, redox activity and mitochondrial functions of three native 1,4-naphtoquinons is of interest. The authors validated with their experiments data from the literature observed on other cell lines or in vivo.

There are a few minor comments and suggestions :

All figures, when printed, are of average to poor quality.

-        Figure legends: please use another police of character or font size to differentiate between the figure legend and the manuscript. At first it seemed that all pertinent information was missing when it was just included in the same format as the text and without separation

Page 1, last line, suggest to write « against most of the antitumour drugs » rather than « against the most antitumour drugs »

Page 2, line 5, use « have » instead of « are » when speaking about vaccines against GBM

Page 2, following sentence, not sure the « of » is necessary in « despite of advances »

Page 5, last paragraph, replace « lawsone » instead of « lavsone »

Page 6, part of  the text is in a different font size.

Page 10, start discussion by « in recent decades, studies… » instead of « recent decade studies… »

Page 10, further down, add « treatment » before « higher than 5µM plumbargin… »

There are also some comments to complement the discussion :

Adding juglone as a reference molecule would have been a strong added benefit to discuss the results.

The difference from other findings with lawsone : could it be linked to some kind of cell type specificity ?

As the three molecules are closely related, can the authors offer hypotheses based on structure activity relationships to explain the differences ?

What are circulating doses and structures of the molecules after food consumption or herbal therapy ? How does it compare to the toxic doses presented in the manuscript ?

Author Response

Response to Reviewer 1 Comments

Dear Reviewer,

Thank you very much for the analysis and useful suggestions for the paper.

Point : All figures, when printed, are of average to poor quality.

Response : The quality of figures is improved

Point :  Figure legends: please use another police of character or font size to differentiate between the figure legend and the manuscript. At first it seemed that all pertinent information was missing when it was just included in the same format as the text and without separation

Response : Character and font size of figure legends is changed

Point :  

Page 1, last line, suggest to write « against most of the antitumour drugs » rather than « against the most antitumour drugs »

Page 2, line 5, use « have » instead of « are » when speaking about vaccines against GBM

Page 2, following sentence, not sure the « of » is necessary in « despite of advances »

Page 5, last paragraph, replace « lawsone » instead of « lavsone »

Page 6, part of  the text is in a different font size.

Page 10, start discussion by « in recent decades, studies… » instead of « recent decade studies… »

Page 10, further down, add « treatment » before « higher than 5µM plumbargin… »

 Response : recommended changes are done.

There are also some comments to complement the discussion :

Adding juglone as a reference molecule would have been a strong added benefit to discuss the results.

Response : We added additional information about effects of juglone (Introduction, Results, Discussion sections) and compared with our results.

The difference from other findings with lawsone: could it be linked to some kind of cell type specificity ?

Response : We agree that there might be cell type specific response to lawsone. However, the properties of the compound might also play the role, because there are studies indicating both cytotoxicity of lawsone and cell protecting antioxidant properties of lawsone. We have added this information to the Discussion section.

As the three molecules are closely related, can the authors offer hypotheses based on structure activity relationships to explain the differences ?

Response : The different effects of Lawsone on the ROS generation compared to the structurally similar compounds (plumbagin and menadione) could be determined by the  -OH group at the C2 position. It was demonstrated that naphthoquinone containing -OH groups attached to quinone moiety exhibited higher antioxidant activity as compared to standard [See Ref No 34 in the Manuscript]. Investigations of other polyphenolic compounds showed that free hydroxyl and C2 - C3 double bound in B-ring are consistently associated with higher antioxidant capacity in polyphenols [See Ref No 35 in the Manuscript]. Moreover, the presence of the -OH group, as compared to the presence of -H and -CH3, also determines the stronger antioxidant properties of the compound [Ref. 36 in the Manuscript].

Point :  What are circulating doses and structures of the molecules after food consumption or herbal therapy ? How does it compare to the toxic doses presented in the manuscript ?

Response : In our studies, plumbagin and menadione were used at amounts 1.8 – 42.5 mg/L and 1.7 – 42.5 mg/L, respectively.

The 1,4- naphthoquinones are used in veterinary practice. A dose of 10 mg/kg completely protected ducks against sporozoite-induced Pgallinaceum infection ; an intravenous dose of 3.16 mg/kg suppressed parasitaemia in monkeys infected with Pcynomolgi. LD50 of lapachol, a 1,4-naphthoquinone derivative that is structurally similar to the compounds investigated in our study is reported between 80 mg/kg to 2.4 g/kg [The Essential Guide to Herbal safety, S. Mills and K. Bone, 2004 Elsevier]. The doses are higher than those of plumbagin and menadione and comparable with those of lawsone used in our research.

Studies with naphthochinones and other groups of polyphenol compounds indicate that they may be chemically unstable in the stomach or neutral medium, and may be decomposed by intestinal microflora [Yang,L. 2017; Sankaranarayanan,R. 2019]. Therefore, detailed in vivo stability and bioavailability studies and mechanism of action of 1,4-naphthoquinone metabolites is an important area for future study. 

Reviewer 2 Report

 Comparison of the effect of native 1,4-naphthoquinons plumbagin, menadione and lawsone on viability, redox status and mitochondrial functions of C6 glioblastoma cells

 Daiva Majiene,Jolita Kuseliauskyte, Arturas Stimbirys and Aiste Jekabsone

This revised manuscript collates current information on the links between dietary xenobiotic and colorectal cancer and the potential interactions with gut microbiota.  This version is much improved as it focusses specifically on CRC and food-gut microbiota interactions.

The overall comment are as follows:

1.       Well written and structured manuscript focussing in the anti-carcinogen effects of three purified naphthoquinones.

2.       There are no line numbers which makes reviewing slightly cumbersome; the font sizes seem to be changing throughout the manuscript.

Detailed comments as follows:

Abstract

The word glioblastoma could be added to C6 the first time it is used to convey the use of the test compounds as anti-cancer compounds.

Introduction

Page 2. 1st para.

Please add reference after side effects.

Delete of after despite.

Define ROS when first used in para 4, then write as ROS in para 5 etc

Three different measures used are: cytotoxicity, pro-oxidant/antioxidant effect and mitochondrial activity. The former two are nicely connected in the last 3 lines of para 4 in page 2. However the mitochondrial function is introduced only in the last line of introduction in page 3. Suggest a link be made between all the 3 measures to understand why these parameters are being studied.

Materials and Methods

Full form of DMEM, HBSS, ADP

The concentrations of the compounds used not mentioned anywhere is the methods. What were the controls (besides the solvent control) used for each of these assays? How long were the cells exposed to the treatments?

A reference with the full protocols of all these assay methods will be useful

Measurement medium contains or contained? (For consistency of tenses)

as nmol O/s-1 /mg-1 /1×106 cells/ml)…. Remove that last bracket here

Results

2.1.  The effect of 1, 4-naphthoquinones on C6 cell viability.

Plumbagin or plumbagine? Lavsone or lawsone? Please check for consistency.

Better to use 12.5 µM rather than 12,5, to be consistent with rest of manuscript. Similar issues elsewhere in the manuscript.

How was IC50 calculated?

The graphs in Figure 2 are very unclear. The panels need to be named with A, B, and C mentioned in the legend.

The sentence “C6 cells were treated with 1 – 20 μM plumbagin, 1-25 μM menadione and 10 – 1000 μM lawsone for 24 hours” would be useful in methods section.

In a part of cells while the other part of the cells… Do you mean …induced necrosis in some cells which other cells within the same treatment has viable blue nuclei.

Also looks like “C6 cells were treated with 1 – 20 μM plumbagin, 1-25 μM menadione and 10 – 1000 μM lawsone for 24 hours. Cell viability was assessed using (1) MTT method and (2) double-staining with Hoechst 33342 and propidium iodide. Hoechst 33342 positive cells but lacking PI staining, were considered as viable cells. PI-stained cells were considered as necrotic. Data are presented as means of percentage of the untreated control cells ± SE (n = 5). * p < 0.05 versus control.” are part of Figure 2 legend?

2.2.  The effect of 1,4-naphthoquinones on intracellular ROS concentration in C6 cell culture

Hours or h, good to use the same for Methods and Results, and the SI unit for time is h.

Figure 3- new terms, PL, Mn and LW introduced in Figure 3.

Looks like  the lines “C6 cells were pre-treated with 10 μM DCFH-DA and then were treated with different concentrations (1-25 μM) of plumbagin and menadione and (10 – 1000 μM) lawsone. Fluorescence intensity, which is proportional to intracellular ROS concentration, was detected by a fluorometer at excitation and emission wavelengths of 544 and 590 nm, respectively. Data are presented as means of percentage of control cells ± SE (n = 5).” are part of legend for Figure 3?

2.3.  Cytotoxic effect of plumbagin was partially prevented by ascorbate

Extent rather than extend.

It would be good to do a statistical analysis to confirm the ROS generation correlation with cytotoxicity.

Ascorbate not mentioned in methods. Not clear whether the ascorbate and the plumbagin were added at same time, or whether one chemical preceded the other.

3.4. The effect of 1,4-naphthoquinones on mitochondrial respiration rate in C6 cells.

 How are the anticancer properties of 1,4-naphthoquinone juglone related to modulation of mitochondrial activity? In what way?

What was the uncoupler?

Discussion

Well discussed, but suggest one more grammar check, especially for tense, etc

Supplementary figures S1B

Y axis title on the left is missing.

Author Response

Response to Reviewer 2 Comments

Dear Reviewer,

Thank you very much for the analysis and useful suggestions for the paper.

Abstract

 The word glioblastoma could be added to C6 the first time it is used to convey the use of the test compounds as anti-cancer compounds.

Response : The word “glioblastoma” is added to the first use of C6 in abstract. It is already present at the first use in text.

Introduction

Page 2. 1st para.

Please add reference after side effects.

Delete of after despite.

Define ROS when first used in para 4, then write as ROS in para 5 etc

Response : The three above indicated changes are made.

Three different measures used are: cytotoxicity, pro-oxidant/antioxidant effect and mitochondrial activity. The former two are nicely connected in the last 3 lines of para 4 in page 2. However the mitochondrial function is introduced only in the last line of introduction in page 3. Suggest a link be made between all the 3 measures to understand why these parameters are being studied.

Response : The explanation of the link between cytotoxicity, oxidant activity and mitochondrial function is inserted in the last paragraph of Introduction section.

Materials and Methods

Full form of DMEM, HBSS, ADP

 Response : changes are done.

The concentrations of the compounds used not mentioned anywhere is the methods. What were the controls (besides the solvent control) used for each of these assays? How long were the cells exposed to the treatments?

Response : The concentrations and treatment duration are added to the Methods section.

A reference with the full protocols of all these assay methods will be useful

Response : the references are added.

Measurement medium contains or contained? (For consistency of tenses)

Response : the tense in Materials and Methods are all made in past form.

as nmol O/s-1 /mg-1 /1×106 cells/ml)…. Remove that last bracket here

Response : the change is made.

Results

2.1.  The effect of 1, 4-naphthoquinones on C6 cell viability.

Plumbagin or plumbagine? Lavsone or lawsone? Please check for consistency.

Response : the names of naphthoquinones are corrected.

Better to use 12.5 µM rather than 12,5, to be consistent with rest of manuscript. Similar issues elsewhere in the manuscript.

Response : changes were done.

How was IC50 calculated?

Response : calculation of IC50 is described in Materials and Methods, sub-section 2.3. Cell viability assessment.

The graphs in Figure 2 are very unclear. The panels need to be named with A, B, and C mentioned in the legend.

Response : The quality of the graphs is improved and the panels are indicated as suggested.

The sentence “C6 cells were treated with 1 – 20 μM plumbagin, 1-25 μM menadione and 10 – 1000 μM lawsone for 24 hours” would be useful in methods section.

Response : The sentence is moved to Materials and Methods section as suggested.

In a part of cells while the other part of the cells… Do you mean …induced necrosis in some cells which other cells within the same treatment has viable blue nuclei.

Response : The sentence is changed to “The fluorescent images obtained confirmed the results of MTT assay: lawsone up to 1 mM induced necrosis only in less than half of cells

Also looks like “C6 cells were treated with 1 – 20 μM plumbagin, 1-25 μM menadione and 10 – 1000 μM lawsone for 24 hours. Cell viability was assessed using (1) MTT method and (2) double-staining with Hoechst 33342 and propidium iodide. Hoechst 33342 positive cells but lacking PI staining, were considered as viable cells. PI-stained cells were considered as necrotic. Data are presented as means of percentage of the untreated control cells ± SE (n = 5). * p < 0.05 versus control.” are part of Figure 2 legend?

Response : Yes, indeed, this was the part of Figure 2 legend. The legend is now better separated from other text. 

2.2.  The effect of 1,4-naphthoquinones on intracellular ROS concentration in C6 cell culture

Hours or h, good to use the same for Methods and Results, and the SI unit for time is h.

Figure 3- new terms, PL, Mn and LW introduced in Figure 3.

Response : changes were done.

Looks like  the lines “C6 cells were pre-treated with 10 μM DCFH-DA and then were treated with different concentrations (1-25 μM) of plumbagin and menadione and (10 – 1000 μM) lawsone. Fluorescence intensity, which is proportional to intracellular ROS concentration, was detected by a fluorometer at excitation and emission wavelengths of 544 and 590 nm, respectively. Data are presented as means of percentage of control cells ± SE (n = 5).” are part of legend for Figure 3?

Response : Yes, indeed, this was the part of Figure 3 legend. The legend is now better separated from other text. 

2.3.  Cytotoxic effect of plumbagin was partially prevented by ascorbate

Extent rather than extend.

Response : the suggested change is done.

It would be good to do a statistical analysis to confirm the ROS generation correlation with cytotoxicity.

Response : According to the suggestion, we have calculated the correlation level between cell viability and ROS production. The results are presented below:

plumbagin

r

-0,9865

95% confidence interval

-0,9973 to   -0,9349

R squared

0,9732

P value

P (two-tailed)

<0,0001

P value summary

****

Significant? (alpha = 0.05)

Yes

Number of XY Pairs

9

Menadione

r

-0,9449

95% confidence interval

-0,9886 to   -0,7539

R squared

0,8929

P value

P (two-tailed)

0,0001

P value summary

***

Significant? (alpha = 0.05)

Yes

Number of XY Pairs

9

Lawsone

r0,771195% confidence interval0,2192 to     0,9492R squared0,5946  P value P (two-tailed)0,0150P value summary*Significant? (alpha = 0.05)Yes  Number of XY Pairs9

In summary, the best correlation is for plumbagin, similar but a bit lower is for menadione.

Ascorbate not mentioned in methods. Not clear whether the ascorbate and the plumbagin were added at same time, or whether one chemical preceded the other.

Response : the both compounds were added at the same time; this is indicated in the Materials and Methods section.

3.4. The effect of 1,4-naphthoquinones on mitochondrial respiration rate in C6 cells.

 How are the anticancer properties of 1,4-naphthoquinone juglone related to modulation of mitochondrial activity? In what way?

Response : In the current study, we have not explored the mechanism of juglone anticancer properties, however, Sidlauskas et al. have shown that juglone suppress mitochondrial complex I and decreases ATP synthesis [Sidlauskas,K. 2017].

What was the uncoupler?

Response : The uncoupler used in the study was dinitrophenol. We added this information into Materials and Methods section. Additionally, this information is written in the  legend of Table 1: VDNF – uncoupled respiration rate in the presence  10 – 50 μM DNF;

Discussion

Well discussed, but suggest one more grammar check, especially for tense, etc

Response : Thank you for the note, the Discussion text is revised for grammar.

Supplementary figures S1B

Y axis title on the left is missing.

 Response : The axis title is added.

Reviewer 3 Report

Researchers of this article tried to study effect of 1,4-naphthoquinons plumbagin, menadione and lawsone on viability, redox status and mitochondrial functions of C6 glioblastoma cells. 

Results section 3.1; quality images have to be improved.  Please be consistent with units used for concentration. ug/ML is not same as uM.  Both in texts and images.

Section 3.4 is not clear.  Please rewrite this section of results. Please explain what is purpose of measuring VL and VADP. 

Discussion is not clear in some paragraphs.  Please rewrite the discussion.

Discuss the differential effects of these compounds based on their chemical structure

Please cite this manuscript (Cancers 2019, 11, 427; doi:10.3390/cancers11030427) which explains role of flavanoids  anti-cancer properties in introduction.

Author Response

Response to Reviewer 3 Comments

Dear Reviewer,

Thank you very much for the analysis and useful suggestions for the paper.

Results section 3.1; quality images have to be improved.  Please be consistent with units used for concentration. ug/ML is not same as uM.  Both in texts and images.

 Response: The quality of the images is improved and the units on the micrographs are corrected.

Section 3.4 is not clear.  Please rewrite this section of results. Please explain what is purpose of measuring VL and VADP.

 Response : We have included the justification of VL and VADP evaluation in the section and rewrited the text accordingly.

Mitochondrial respiration was measured in non-phosphorylating state, without external addition of ADP, and in the presence of 1mM ADP that causes maximal stimulation of phosphorylation and subsequent activation of respiratory chain. Non-phosphorylating respiration rate is named VL (L for “leak”) because it is caused only by proton leak across the inner mitochondrial membrane and is an indicator of the membrane permeability. ADP-stimulated respiration rate is named VADP and provides information about the maximal respiratory capacity of oxidative-phosphorylation system  [Ref 20 in the Manuscript].

Discussion is not clear in some paragraphs.  Please rewrite the discussion.

 Response: The discussion is restructured and supplemented with additional information.

Discuss the differential effects of these compounds based on their chemical structure. Please cite this manuscript (Cancers 2019, 11, 427; doi:10.3390/cancers11030427) which explains role of flavanoids anti-cancer properties in introduction.

Response : studies with naphthoquinones and other groups of polyphenol compounds indicate that they may be chemically unstable in the gastrointestinal medium, and may be decomposed by intestinal microflora [See Ref No 25 and 26 in the Manuscript]. Therefore, detailed in vivo stability and bioavailability studies and mechanism of action of 1,4-naphthoquinone metabolites on gastrointestinal tissue is an important area for future study. 

Round 2

Reviewer 3 Report

Please accept this manuscript.